# Promises and Pitfalls of Next-Generation Treg Adoptive Immunotherapy

**DOI:** 10.3390/cancers15245877

**Published:** 2023-12-17

**Authors:** Panayiota Christofi, Chrysoula Pantazi, Nikoleta Psatha, Ioanna Sakellari, Evangelia Yannaki, Anastasia Papadopoulou

**Affiliations:** 1Gene and Cell Therapy Center, Hematopoietic Cell Transplantation Unit, Hematology Department, George Papanikolaou Hospital, 57010 Thessaloniki, Greece; panayiota2ch@hotmail.com (P.C.); chrysa_1@hotmail.gr (C.P.); ioannamarilena@gmail.com (I.S.); eyannaki@uw.edu (E.Y.); 2University General Hospital of Patras, 26504 Rio, Greece; 3Department of Genetics, Development and Molecular Biology, School of Biology, Aristotle University of Thessaloniki, 54124 Thessaloniki, Greece; npsatha@bio.auth.gr; 4Institute of Applied Biosciences (INAB), Centre for Research and Technology Hellas (CERTH), 57001 Thessaloniki, Greece; 5Department of Medicine, University of Washington, Seattle, WA 98195-7710, USA

**Keywords:** regulatory T cells, adoptive immunotherapy, autoimmune diseases, transplantation, CAR tregs

## Abstract

**Simple Summary:**

Adoptive immunotherapy has emerged as an effective alternative of mounting impact to the current standard of care in cancer, viral infections, and recently, autoimmunity. Key players in maintaining immune homeostasis are the regulatory T cells (Tregs), a major immunosuppressive cell subset and, therefore, an attractive candidate for the cellular therapy of autoimmune disorders or allo-responses in the transplantation setting. Notwithstanding the safety and tolerability of Tregs in early trials, their efficacy remains rather ill-defined, being limited by poor persistence and a lack of specificity, thus hindering widespread clinical application. However, the better biological understanding of in vivo Treg performance and the recent advances in genetic engineering have led to the next-generation Treg immunotherapy era, enabling the introduction of new features in Tregs and generating more potent and targeted Treg cellular therapies. In this review, we discuss the current achievements and existing challenges towards clinically translating Tregs into a living drug therapy for a variety of inflammatory conditions.

**Abstract:**

Regulatory T cells (Tregs) are fundamental to maintaining immune homeostasis by inhibiting immune responses to self-antigens and preventing the excessive activation of the immune system. Their functions extend beyond immune surveillance and subpopulations of tissue-resident Treg cells can also facilitate tissue repair and homeostasis. The unique ability to regulate aberrant immune responses has generated the concept of harnessing Tregs as a new cellular immunotherapy approach for reshaping undesired immune reactions in autoimmune diseases and allo-responses in transplantation to ultimately re-establish tolerance. However, a number of issues limit the broad clinical applicability of Treg adoptive immunotherapy, including the lack of antigen specificity, heterogeneity within the Treg population, poor persistence, functional Treg impairment in disease states, and in vivo plasticity that results in the loss of suppressive function. Although the early-phase clinical trials of Treg cell therapy have shown the feasibility and tolerability of the approach in several conditions, its efficacy has remained questionable. Leveraging the smart tools and platforms that have been successfully developed for primary T cell engineering in cancer, the field has now shifted towards “next-generation” adoptive Treg immunotherapy, where genetically modified Treg products with improved characteristics are being generated, as regards antigen specificity, function, persistence, and immunogenicity. Here, we review the state of the art on Treg adoptive immunotherapy and progress beyond it, while critically evaluating the hurdles and opportunities towards the materialization of Tregs as a living drug therapy for various inflammation states and the broad clinical translation of Treg therapeutics.

## 1. Introduction

Regulatory T cells (Tregs), consist of a small, albeit critical for maintaining immune equilibrium, fraction of T cells, that prevent or dampen immune responses to self-antigens, thus preserving self-tolerance while suppressing excessive immune activation to non-self-antigens ([1,2,3]). According to the recommendations on Treg cell nomenclature [4], Tregs can be classified based on their origin of differentiation as (i) emerging de novo in the thymus, thus bearing a T cell receptor (TCR) with specificity towards self-antigens [thymus-derived Tregs [5], tTregs, previously called natural Tregs (nTregs)]; (ii) differentiating in the periphery, therefore having a non-self-antigen-specific TCR [peripherally derived Tregs, pTregs, previously known as induced or adaptive Tregs (iTregs or aTregs)]; and (iii) being generated ex vivo (in vitro-induced Tregs, iTregs) and clearly distinguished from the in vivo-generated Tregs. Despite their heterogeneity, there is a lack of specific markers distinguishing human tTregs from pTregs. The classically defined Tregs are CD4+ cells, constitutively expressing high levels of CD25 (interleukin-2 receptor alpha chain, IL-2Rα) and the transcription factor forkhead box P3 (Foxp3). These markers are expressed by the majority of tTregs and also a subpopulation of pTregs [6]. Two additional Foxp3- pTreg subsets, T-helper 3 (Th3) and type-1 Treg (Tr1), the suppressive functions of which rely on transcription growth factor beta (TGF-β) and interleukin-10 (IL-10) secretion, respectively, have also been well-described [7,8]. To date, there is also convincing, albeit substantially less than for CD4+ Tregs, evidence for the existence of CD8+ Tregs with properties similar to their CD4+ counterparts [9]. Table 1 outlines the main extracellular markers and transcription factors expressed by various Treg subtypes, along with their respective mechanism of immunosuppressive action.

Tregs reshape immune responses with precision, executing their regulatory function in a sophisticated and tailored manner as opposed to the conventional, general immunosuppressive approaches. This precise immune regulation, particularly in contexts like autoimmunity and transplantation is of highest importance. Strategies boosting polyclonal Treg numbers and function in vivo by the administration of Treg-promoting proteins or pharmacological agents such as interleukin-2 (IL-2) [10,11,12], anti-IL-2 complexes [13], intravenous immunoglobulin alone or in combination with rapamycin [14,15,16,17], antibody-mediated agonistic stimulation of tumor necrosis factor superfamily receptor 25 (TNFRSF25), and cytokine-targeted antibodies, which modify the pro-inflammatory environment rescuing Treg function [18,19,20,21,22], have enhanced the in vivo tolerance in preclinical studies and early clinical trials.

Adoptive immunotherapy with Tregs, which includes the isolation and ex vivo expansion of autologous Tregs, has emerged as an attractive therapeutic option to restore the immune balance in autoimmunity and transplantation. Multiple studies using the adoptive transfer of ex vivo-expanded polyclonal Tregs have demonstrated significant potential for inducing tolerance and preventing graft rejection following solid organ transplantation [23], as well as treating autoimmune-mediated diseases, including type 1 diabetes (T1D) [24], rheumatoid arthritis [25], multiple sclerosis [26], and systemic lupus erythematosus (SLE), or graft-versus-host disease (GvHD) in the allogeneic hematopoietic cell transplantation setting, and recently, coronavirus disease 2019 (COVID-19) [27,28,29]. However, increasing evidence suggests that Tregs are functionally impaired in patients with autoimmune diseases and transplant recipients, due to the instability of Foxp3 expression, impaired suppressive function, decreased migratory capacity, and increased apoptosis [30,31,32]. The early results of the first clinical trials, although promising, have questioned the efficacy of Tregs, as only modest clinical responses were achieved. In this review, we focus on the obstacles limiting the clinical utility of Treg adoptive immunotherapy in the context of autoimmunity and transplantation and discuss strategies to overcome these impediments and improve the outcomes with Treg cell therapy (Table 2).

**Table 1 cancers-15-05877-t001:** Extracellular markers, transcription factors, and mechanisms of actions of various Treg subtypes.

T_reg_ Subset	Origin	Markers	Transcription Factors	Suppressive Mechanism	References
tT_regs_	Generating in the thymus	CD4^+^, CD25^hi^, CD27^lo^, CTLA-4^+^, LAG-3^+^, TIGIT^+^, TIM-3^+^, PD-1+	FOXP3pos	Cell-contact-dependent immunosuppression via receptors like CTLA-4 and PD-1	[33]
pT_regs_	Differentiating from peripheral naive CD4+ T cells	CD4^+^, CD25^hi^, CD27^lo^, CTLA-4+, LAG-3+, TIGIT+, Tim-3+, PD-1+	FOXP3pos	Inhibitory function via soluble factors such as TGF-β1 and IL-10	[33]
Tr1 T_regs_	Differentiating from peripheral naive CD4+ T cells	CD4^+^, CD25, CD49b^+^, LAG-3^+^ [3]	Tbet, Blimp-1, FOXP3neg [1]	Inhibitory function via IL-10 production	[34,35]
Th3 T_regs_	Differentiating from peripheral naive CD4+ T cells	CD4^+^, CD25^+^, CD69^+^, LAP^+^	TGF-β, FOXP3neg	Inhibitory function via TGF-β production	[36]
CD8^+^ T_regs_	Differentiating from peripheral naive CD8+ T cells	CD8^+^, CD25^+^, CD122+, CD49d+	FOXP3pos, Eomes, Helios, TGF-β	TGF-dependent control of Helios and homeostatic cytokine IL-15 [4]	[37,38]

tTregs: thymus-derived Tregs, pTregs: peripherally derived Tregs, Tr1 Tregs: Type 1 Tregs, Th3 Tregs: T helper T cells, CTLA-4: Cytotoxic T-lymphocyte-associated protein 4, LAG-3: Lymphocyte-activation gene 3, TIGIT: T-cell immunoreceptor with immunoglobulin and immunoreceptor tyrosine-based inhibitory motif domains, TIM3: T-cell immunoglobulin and mucin domain-containing protein 3, PD-1: Programmed cell death protein 1, FOXP3: Forkhead box P3, TGF-β: Transforming Growth Factor β 1, IL-10: Interleukin-10, Tbet: T-box expressed in T cells, Blimp-1: B lymphocyte-induced maturation protein 1, LAP: Latency-associated peptide positive, Eomes: Eomesodermin, IL-15: Interleukin-15.

**Table 2 cancers-15-05877-t002:** Clinical trials of adoptive immunotherapy with Tregs.

Cell Product	Source	Disease	Treg Manufacturing	Study Phase	Patients	Safety	Efficacy	Trial ID	References
Tregs for autoimmune diseases									
Polyclonal Tregs	Autologous	T1D	Isolation/enrichment and ex vivo expansion	I	12	No AEs	8/12 clinical remission	ISRCTN06128462	Marek-Trzonkowska et al., 2014 [39]
Polyclonal Tregs	Autologous	T1D	Isolation/enrichment and ex vivo expansion	I	14	Well-tolerated. No cell therapy-related high-grade AEs	Not powered to detect improvement in metabolic function	NCT01210664	Bluestone et al., 2015 [24]
Polyclonal Tregs	Autologous	T1D	Ex vivo expansion	II, randomized placebo-controlled double blind	110	Well-tolerated	No improvement in the preservation of C-peptide levels vs. placebo	NCT02691247	Caladrius Biosciences, 2019 [40]
Polyclonal Tregs	UCB	T1D	Isolation/enrichment and ex vivo expansion	I/II, randomized, parallel assignment, open label	Recruiting			NCT02932826	
Combinational: polyclonal Tregs + low-dose IL-2	Autologous	T1D	Isolation/enrichment and ex vivo expansion	I	7	Off-target effect of low-dose IL-2 (dramatic reduction in C-peptide production and potential shift of the immune balance toward activation rather than tolerance)—terminated	No preservation or improvement of C-peptide production	NCT02772679	Dong et al., 2021 [41]
Combinational: polyclonal Tregs + anti-CD20	Autologous	T1D	Isolation/enrichment and ex vivo expansion	I/II, randomized, three-arm, open-label, single-blinded	36 paediatric (Tregs only n = 13, Tregs + rituximab n = 12, control n = 11)	AEs in 80% of pts (combined group and Tregs only group). AEs, such as infections, needed special surveillance	Tregs+anti-CD20 were superior than Tregs in controlling recent-onset T1DM regarding C-peptide levels and remission	TregVAC2.0; EudraCT: 2014-004319-35	Zieliński et al., 2022 [42]
Combinational: polyclonal Tregs + Liraglutide	UCB	T1D	Isolation/enrichment and ex vivo expansion	I/II, randomized, parallel assignment, open label	Recruiting			NCT03011021	
Polyclonal Tregs	Autologous	MS	Tregs for iv: isolation/enrichment and ex vivo expansion Tregs for IT: isolation/enrichment	1b/2a (randomized to iv or IT Treg administration)	14 (iv n = 11, IT n = 3)	No severe AEs	5/11 relapses (iv-treated), 0/3 relapses (IT-treated). The statistical results may be underpowered due to the low number of patients	EudraCT 2014–004320-22	Chwojnick et al., 2021 [43]
Polyclonal Tregs	Autologous	Autoimmune hepatitis	Isolation/enrichment and ex vivo expansion	I/II	Unknown status			NCT02704338	
Polyclonal Tregs	Autologous	Active cutaneous lupus	Isolation/enrichment and ex vivo expansion	I	1	Terminated due to participant recruitment feasibility	Stable clinical status	NCT02428309	Dall’Era et al., 2019 [28]
Polyclonal Tregs	Autologous	Active Pemphigus	Isolation/enrichment and ex vivo expansion	I	5	Terminated due to recruitment issues and the impact of the coronavirus infectious disease 19 (COVID-19) pandemic		NCT03239470	
Ag-specific, ovalbumin-specific type 1 Tregs (ova-Tregs)	Autologous	Refractory Crohn’s disease	Isolation/enrichment and ex vivo expansion	I/IIa	29 enrolled, 20 treated	Well-tolerated, good safety profile for this small patient cohort—significant AEs primarily related to the gastrointestinal system and the underlying CD	8/20 (40%) total clinical improvement and 6/8 (75%) clinical response in the low-dose group (reducing dose-dependent efficacy)	NCT02327221	Desreumaux et al., 2012 [44]
Polyclonal Tregs	Autologous	Crohn’s disease	Isolation/enrichment and ex vivo expansion	I	Recruiting			NCT03185000	
Tregs for solid organ transplantation									
Donor-alloantigen-specific Tregs	Autologous	Liver transplantation	Ex vivo expansion	I/IIa	10	Good safety profile	10/10 normal graft function and histology. 7/10 successful cessation of immunosuppressive drugs. 3/10 required conventional low-dose immunotherapy	n/a	Todo et al., 2016 [45]
Donor-alloantigen-specific Tregs	Autologous	Liver transplantation	Ex vivo expansion	I	15	Terminated as it could not be completed within the grant timeline		NCT02188719 (darTregs) in Liver Transplantation (deLTa)	
Donor-alloantigen-specific Tregs	Autologous	Liver transplantation	Isolation/enrichment and ex vivo expansion	I	Unknown status			NCT01624077	
Donor-alloantigen-specific Tregs	Autologous	Liver transplantation	Isolation/enrichment and ex vivo expansion	I/II	15	Not sufficiently powered to assess safety or efficacy (only n = 5 finally received Tregs)		NCT02474199 (ARTEMIS)	Tang Q et al., 2022 [46]
Polyclonal Tregs	Autologous	Liver transplantation	Ex vivo expansion	I/II	9 (3 received 10^6^ Tregs/kg, 6 received 4.5 × 10^6^ Tregs/kg)	Good safety profile	6/6 of the high-dose-treated demonstrated reduced donor-specific T cell responses	NCT02166177 (ThRIL)	Sánchez-Fueyo et al., 2020 [47]
Donor-alloantigen-specific Tregs	Autologous	Liver transplantation	Isolation/enrichment and ex vivo expansion	I/II	Active, not recruiting			NCT03577431(ITN073ST)	
HLA-A∗02-CAR Tregs	Autologous	Liver transplantation	Ex vivo expansion and genetic engineering	I/II	Recruiting			NCT05234190 (LIBERATE)	
HLA-A∗02-CAR Tregs	Autologous	Kidney transplantation	Ex vivo expansion and genetic engineering	I/II	Recruiting			NCT04817774 (Steadfast)	Schreeb et al., 2022 [48]
Polyclonal Tregs	Autologous	Kidney transplantation	Isolation/enrichment and ex vivo expansion	I	3	Well-tolerated	2/3 improvement in follow-up biopsies	NCT02088931 (TASKp pilot trial)	Chandran et al., 2017 [49]
Polyclonal Tregs	Autologous	Kidney transplantation	Ex vivo expansion	I	9	Good safety profile	All pts survived for at least 2 years	NCT02145325 (TRACT trial)	Mathew et al., 2018 [50]
Polyclonal Tregs	Autologous	Kidney transplantation	Isolation/enrichment and ex vivo expansion	n/a	Recruiting			NCT03284242	
Combinational: polyclonal Tregs+ donor bone marrow cells + Tocilizumab	Autologous	Kidney transplantation	Isolation/enrichment and ex vivo expansion	I/IIa	Active, not recruiting			NCT03867617 (Trex001)	Oberbauer et al., 2021 [51]
Polyclonal Tregs	Autologous	Kidney transplantation	Isolation/enrichment and ex vivo expansion	I/II	Unknown status			NCT01446484 (RSMU-001)	
Polyclonal vs. donor-specific Tregs	Autologous	Kidney transplantation	Ex vivo expansion	I/II randomized open-label	n/a	Completed. No results posted yet		NCT02711826 (TASK, CTOT-21)	
Polyclonal and donor-antigen reactive Tregs, tolerogenic dendritic cell and regulatory macrophage cells	Autologous	Kidney transplantation	Isolation/enrichment and/or ex vivo expansion	7 phase I/II trials	66 cell-treated group vs. 38 reference-group	Good safety profile	Lower infection rates; rates of biopsy-confirmed acute rejection (BCAR) comparable between the standard immunosuppressive group and the cell-based therapy group. Successfully weaned off immunosuppression within the first year post-transplantation to monotherapy in nearly all cell-treated patients	NCT02371434, NCT02129881 (polyclonal Treg), NCT02244801, NCT02091232 (donor-antigen reactive Treg), NCT02252055 (tolerogenic dendritic cell), NCT02085629 (regulatory mac rophage cell), NCT01656135 (reference group) (ONE study)	Sawitzki et al., 2020 [52]
Combinational: total lymphoid irradiation (TLI), total body irradiation (TBI), anti-thymocyte globulin (ATG), donor HSCs and polyclonal Tregs	Autologous	Kidney transplantation	Ex vivo expansion	I	Recruiting			NCT03943238	
Polyclonal Tregs	Autologous	Kidney transplantation	Ex vivo expansion	IIb, randomized	Recruiting			ISRCTN11038572 (Two study)	Brook et al., 2022 [53]
Polyclonal Tregs	Autologous	Heart transplantation	Isolation/enrichment and ex vivo expansion	I/II, randomized	Recruiting			NCT04924491 (THYTECH)	Bernaldo-de-Quirós et al., 2022 [54]
Polyclonal Tregs	Autologous	Islet transplantation	Isolation/enrichment and ex vivo expansion	I	Active, not recruiting			NCT03444064	
Tregs for COVID-19									
Polyclonal Tregs	Allogeneic, UCB	COVID-19	Isolation/enrichment and ex vivo expansion	I, randomized, double-blinded, placebo-controlled clinical trial	45 (15 pts placebo, 15 pts 100 × 10^6^ Tregs, 15 pts 300 × 10^6^, 3 doses Tregs)	Good safety profile	No definitive conclusions with respect to efficacy due to to the low number of patients	NCT04468971	Gladstone et al., 2023 [55]
Tregs for GvHD									
Polyclonal HLA-G + induced T-regulatory cells (iG-Tregs)	Allogeneic, HLA-identical sibling donor-derived	GvHD prophylaxis	Ex vivo expansion	I/II	Recruiting			EUDRACT-2021-006367-26	Lysandrou et al., 2023 [56]
Polyclonal Tregs	Allogeneic, HLA-matched sibling donor-derived	GvHD treatment	Isolation/enrichment and ex vivo expansion	I	2	Temporal control of grade IV acute GvHD refractory to all other immunosuppressants used/significant alleviation of chronic GvHD accompanied by reduced pharmacologic immunosuppression		NKEBN/458-310/2008	Trzonkowski et al., 2009 [57]
Polyclonal Tregs	Allogeneic, partially HLA-matched third UCB	GvHD prophylaxis	Isolation/enrichment	I	23	No infusional toxicities	No adverse effect in terms of infection, relapse, or early mortality/decreased incidence of grade II–IV acute GVHD vs. identically treated historical controls	NCT00602693	Brunstein et al., 2011 [58]
Polyclonal Tregs	Allogeneic, HSC donor-derived	Severe refractory GvHD treatment	Isolation/enrichment	I/II	Completed. No results posted yet			NCT02749084	
Polyclonal Tregs	Allogeneic, UCB donor-derived	GvHD prophylaxis	Not specified	II	3	2/3 AEs/Treg-cell infusion toxicity	2/3 grade II-IV acute GvHD; 1/3 bacterial infection; 2/3 viral infection. Terminated due to the consideration of new technology for the product	NCT02991898	
Polyclonal Tregs	Allogeneic, HLA-matched sibling donor-derived	Steroid dependent/refractory chronic GvHD treatment	Not specified	I	Completed. No results posted yet			NCT01911039	
Combinational: polyclonal Tregs + low-dose IL-2	Allogeneic, HSC donor-derived	Steroid refractory chronic GvHD treatment	Isolation/enrichment	I	25	Good safety profile	5/25 (20%) PR; 10/25 (40%) stable disease	NCT01937468	Whangbo et al., 2022 [59]
Polyclonal Tregs	Allogeneic, HSC donor-derived	Steroid refractory chronic GvHD treatment	Isolation/enrichment	I/II	Unknown status			NCT02385019	
Combinational: polyclonal Tregs + IL-2 + rapamycin	Allogeneic, HSC donor-derived	Chronic GvHD treatment	Isolation/enrichment	II	Teriminated due to slow recruitment			NCT01903473	
Combinational: polyclonal Tregs + Tcon	Allogeneic, HSC donor-derived	GvHD prophylaxis + GvL augmentation in pts with high-risk hematological malignancies undergoing allogeneic myeloablative (MA) HCT with a T cell-depleted graft	Isolation/enrichment	I/II	Interim results: 12 (initial group: 5 pts with frozen Tregs, modified groupI:7 pts with fresh Tregs and single-agent GVHD prophylaxis)	No infusion reaction	Initial group: 2/5 grade II GvHD; modified group: 0/7 GvHD	NCT01660607	Meyer et al., 2019 [60]
Polyclonal, fucosylated Tregs	Allogeneic UCB-derived	GvHD prophylaxis	Isolation/enrichment and ex vivo expansion	I	5	No infusion reaction	5/5 ≥grade II acute GVHD. No longterm complications for 4/5 alive pts	NCT02423915	Kellner et al., 2018 [61]
Alloantigen-specific Tr1 cells	Allogeneic,HSC donor-derived	GvHD prophylaxis	Ex vivo expansion	I	3 (preliminary results)	No AEs post infusion	3/3 alive, disease-free and acute GvHD-free at 1 year post-HCT	NCT03198234	Chen et al., 2021 [62]
Polyclonal Tregs	Allogeneic,HSC donor-derived	GvHD prophylaxis	Isolation/enrichment and ex vivo expansion	I	14	No severe infusional toxicities	Pts receiving sirolimus/MMF: 2/2 grade III acute GvHD pts receiving CSA/MMF: 5/12 acute GvHD grade II-III, 6/12 chronic GvHD	NCT01634217	MacMillan et al., 2021 [63]
Polyclonal Tregs	Allogeneic, HSC donor-derived	Steroid refractory chronic GvHD treatment	Isolation/enrichment	II	Recruiting			NCT05095649	
CD6-CAR Tregs	Allogeneic,HSC donor-derived	Chronic GvHD treatment	Ex vivo expansion and genetic engineering	I	Not yet recruiting			NCT05993611	

Treg: regulatory T cells. T1D: type 1 diabetes. AEs: adverse events. MS: multiple sclerosis. Ag-specific: antigen-specific. UCB: umbilical cord blood. Tr1 cells: type 1 regulatory T cells. GvHD: graft versus host disease. Pt: patient; PR: partial response. IL: interleukin. IT: intrathecal. MMF: mycophenolate mofetil. CSA: cyclosporine. HSC: hematopoietic stem cells.

## 2. Specificity of Tolerance

Although there are a number of ongoing clinical trials for autoimmune disorders using polyclonal Tregs (NCT0469123, NCT02772679), the use of polyclonal Tregs, exhibiting a plethora of different TCR specificities, has been hampered by fundamental limitations including the lack of antigen specificity, the heterogeneity of the cell population, and an exhausted Treg phenotype during ex vivo expansion. The suppressive activity of polyclonal Tregs is shaped after ex vivo Ag activation via their TCR, prior to adoptive transfer; however, once stimulated, activated Tregs exert non-specific suppression in an Ag-independent manner post in vivo administration. Such generalized immunosuppression may enhance the risk of opportunistic infections or tumor growth in transplanted or tumor-bearing hosts, respectively [64,65,66].

In addition, as polyclonal Tregs do not consist of a homogeneous population, an infusion of large numbers of cells is required for clinical benefit, yet at the expense of nonspecific immune suppression. At last, the observed loss of the Treg phenotype and attenuation of their immunosuppressive function upon repetitive polyclonal TCR and CD28 co-receptor-mediated stimulation during ex vivo expansion [67] further limits polyclonal Treg potency. To overcome these hurdles, many groups are engaged in pursuing alternatives to polyclonal Tregs.

### Generating Antigen-Specific Tregs to Overcome Polyclonal Treg Limitations

In contrast to polyclonal Tregs, an enriched population of antigen-specific Tregs, which would mainly migrate towards the sites of cognate antigen presentation, may provide the advantage of on-target specificity, without global immunosuppression. In addition, due to the enhanced trafficking to, and the targeted immunosuppression in diseased tissues, lower numbers of antigen-specific Tregs are required for clinically relevant outcomes over their unselected, polyclonal counterparts [68]. Thus, immunotherapy with antigen-specific Tregs may be both safer and more potent than Tregs of a polyclonal TCR repertoire in inducing immune tolerance in a disease-specific manner.

Indeed, numerous studies attest to the clinical superiority of antigen- or alloantigen-specific Tregs, showcasing their increased suppressive efficacy, improved migration patterns to the target tissue, and limited off-target effects as compared to polyclonal Tregs [68,69,70,71,72,73]. Nevertheless, the ex vivo, large-scale expansion of disease-relevant, antigen-specific Tregs is hampered mainly by their low frequency in peripheral blood (merely 1–3% of the circulating CD4+ T cells [74]). Thus, expanding sufficient doses for clinical use requires the implementation of prolonged, labor-intensive, and costly protocols, largely yielding Treg products of suboptimal quality with compromised Treg suppressive ability. Hence, current efforts are focusing on the ex vivo manufacturing of antigen-specific Tregs, either by converting antigen-specific conventional T cells into FOXP3+ cells with suppressive function or redirecting the specificity of polyclonal Treg cells by genetic engineering to express a synthetic antigen receptor that recognizes a disease-relevant antigen (Figure 1) [75].

The first approach was initially reported by Stephens et al., who converted naive CD4+ Foxp3− T cells specific for a naturally expressed autoantigen (H+/K+ ATPase) into self-antigen-specific Foxp3+ Tregs, by stimulation in the presence of TGF-β [76]. Those naïve organ-specific Tregs proved effective at preventing autoimmunity in a murine model of autoimmune gastritis. More recently, Akamatsu et al. showed that epigenetic modification induced by the chemical inhibition of the cyclin-dependent kinase 8 (CDK8) and CDK19, enabled the conversion of antigen-specific effector/memory T cells into Foxp3+ cells [75]. The in vivo inhibition of CDK8/19 generated functionally stable FoxP3+Tregs, capable of suppressing immune responses in mouse models of multiple sclerosis, allergy, and diabetes.

The second approach of converting primary T cells into antigen-specific Tregs towards targeted immune suppression involves a plethora of genetic engineering technologies, including retro/lenti-viral transduction or non-viral transfection methods, such as DNA-based transposons, CRISPR/Cas9 technology, or the direct transfer of in vitro transcribed messenger RNA (mRNA), allowing the introduction and ultimately the expression of either artificial TCRs (TCR-Tregs) or chimeric antigen receptors (CAR-Tregs) into Tregs [77]) (Figure 1).

TCRs isolated from islet-specific human T cells and delivered into polyclonal Tregs provided the proof of concept for the development of islet-specific Treg therapies for the effective treatment of Type 1 diabetes (T1D) [78]. Similarly, Tregs expressing a myelin basic protein-specific (MBP) TCR ameliorated the severity of disease in mouse models of multiple sclerosis [79,80]. Given that in many autoimmune diseases, the causative antigen is often not defined, Wright et al. leveraged the ability of Tregs to also promote bystander suppression once activated—in other words, the ability of activated Tregs to recognize unrelated antigens in the local microenvironment and create a regulatory milieu suppressing conventional T cells (Tcons), independently of antigen specificity [66,81]—and explored whether Tregs, transduced with a TCR specific to a disease-unrelated antigen, could direct their suppressive function to selective sites in vivo and ameliorate the autoimmune disease. Indeed, adoptive therapy with ovalbumin-specific TCR-Tregs in an established arthritis model resulted in the amelioration of arthritis via bystander suppressive pathways, in the absence of cognate recognition of disease-initiating antigen [82], suggesting a clear clinical benefit by tissue-specific TCR Tregs in the treatment of autoimmune diseases even when the disease-causing autoantigens remain unknown. Beyond autoimmunity, the combination of TCR-Tregs specific for allogeneic major histocompatibility complex (MHC) class II molecules with short-term adjunctive immunosuppression, favored transplantation tolerance in mice, implying clinical potential for the administration of Tregs bearing a TCR specific for donor antigens. Likewise, Tregs derived from TCR transgenic mice targeting the minor histocompatibility antigen (miHAg) HY, which is expressed solely in male mice, were highly effective in controlling GvHD in an antigen-dependent manner while sparing the GVL effect in haploidentical and miHAg-mismatched murine bone marrow transplantation models [83,84].

Tregs have been also engineered to express CARs towards suppressing Ag-specific immune responses in various diseases and several proof-of-concept studies demonstrated the utility of CAR Tregs in the setting of autoimmunity and transplantation. CARs recognize a specific antigen in an MHC-independent mode via an extracellular fusion protein of the variable regions of the heavy and light chain of a specific immunoglobulin which is linked via a transmembrane domain to the intracellular signaling domain CD3z, allowing for T cell activation upon antigen encounter.

Elinav et al. first reported that the adoptive transfer of CAR-Tregs targeting the colitis-associated antigen 2,4,6-trinitrophenol (TNP) for the treatment of induced colitis [85] ameliorated experimental colitis over wild type Tregs, thus paving the way for the treatment of inflammatory diseases using CAR-Tregs. Following this study, many groups reported data suggesting the preliminary success of CAR-Tregs in experimental models of autoimmune and chronic inflammatory diseases, including inflammatory bowel disease [86,87,88], multiple sclerosis [89], T1D [90], asthma [91], and hemophilia [92]. Apart from autoantigens, Tregs can be also engineered to suppress alloimmune responses and promote transplantation tolerance via the CAR targeting of donor MHC molecules. CAR-Tregs targeting human leukocyte antigen (HLA)-A2, the most common, frequently mismatched, antigen in transplantation, have been shown to efficiently prevent lethal GvHD [93,94], and induce graft-specific tolerance after pancreatic islet, skin, or heart graft in mouse models [95,96,97,98,99]. These promising findings led to the authorization of the first-in-human trials assessing the safety and tolerability of autologous CAR-Tregs in HLA-A*02-negative recipients receiving renal and liver transplants, from an HLA-A*02-positive donor (NCT04817774 and NCT05234190, respectively [48]). Lastly, CD19 has also been targeted by CAR-Tregs, as autoantibodies secreted from B cells are thought to induce various autoimmune diseases [100]. In a xenograft mouse model, CD19-CAR-Tregs showed the efficient suppression of IgG antibody by B cells and the differentiation of B cells, without inducing GvHD, providing a novel strategy to treat autoantibody-mediated autoimmune diseases [101]. In this context, conventional CD19-CAR T cells, yet not CD19-CAR Treg cells, have been shown to successfully treat refractory SLE in humans [102].

Technological advances in engineering Tcons for cancer therapy have also inspired their integration into Treg immunotherapy. Biswas’s group redirected the specificity of Tregs towards the coagulation factor (F)VIII, either by delivering a high-affinity CAR (second generation CAR) or a TCR fusion construct (TRuC) synthesized by fusing the FVIII single-chain variable fragment (scFv) to the TCRε subunit, enabling T cell activation independently of a peptide–MHC complex, and compared those two Treg cell products side by side [103]. Surprisingly, CAR-Treg engagement induced a robust effector phenotype resulting in the loss of their suppressive function. In contrast, TruC Tregs delivered controlled antigen-specific signaling via the engagement of the entire TCR complex and successfully suppressed the FVIII-specific antibody response, implicating that cellular therapies employing engineered receptor Tregs may require the fine-tuning of activation thresholds to optimize their suppressive performance. CARs engineered with a modular approach are called UniCARs and have also been employed in CAR-Treg therapy. In Uni CARs, the antigen recognition domain is split from the signaling domain of a conventional CAR. This CAR system contains a signaling module that binds to a specific epitope on a switching/targeting module, which is a bispecific fusion molecule harboring one binding domain directed against a tumor-associated Ag and an epitope specifically recognized by the UniCAR. Hence, UniCAR T cells are switchable and remain dormant until they encounter the targeting module and are cross-linked to target cells. The target antigen can be readily adjusted if needed, by targeting module exchange, without the requirement of re-engineering the CAR T cells [104]. These CARs can be therefore applied universally. The rationale behind UniCAR Tregs has been tested to a limited extent thus far. UniCAR Tregs have been generated from patients with autoimmune or inflammatory diseases or healthy volunteers and when infused in mouse models, they were localized at specific sites and mitigated inflammatory or allograft responses in a spatiotemporal manner [95,105]. Although limited, these findings provide evidence for the feasibility of UniCAR adaptation in Tregs. To our knowledge, UniCAR Tregs have not yet been tested in the clinical setting. However, immunotherapy with UniCAR Tregs with an on/off switchable potential may offer a safer approach, enabling a flexible, albeit precise, modular targeting for Treg adoptive immunotherapy of inflammation-related diseases including GvHD, autoimmunity, or transplant rejection. In another context, De Paula Pohl et al. developed a CAR-analogous, chimeric B-cell antibody receptor, called BAR, containing the immunodominant A2 domain of FVIII to generate BAR-Tregs targeting FVIII-specific B cells which are responsible for persistent anti-FVIII neutralizing antibodies (inhibitors) in hemophilia A patients [106]. This in vitro study demonstrated that only A2-FVIII domain-expressing BAR Tregs, but not A2-BAR Tcons, could efficiently target and suppress FVIII-specific memory B cells. Other approaches driving the antigen specificity of Tcons, such as third- and fourth-generation CARs, and TCR-like CARs (CAR T cells with a TCR-like antibody) could also be applied to Treg-based therapies [107,108,109,110].

## 3. Treg Functional Stability versus Plasticity

The T cell phenotype is inseparably linked to its activity; thus, any phenotypic alteration of T cells will significantly skew their function. Studies over the past few decades have established that within an inflammatory niche, some Tregs present lineage instability, losing the expression of FoxP3, the master regulator of Treg cell differentiation and function [6,111], and thus, the ability to sustain repressor functions or/and an unexpected plasticity enabling rapid cell fate conversion from a suppressive to an active, effector T cell immune phenotype and function (as reviewed in [112,113,114,115,116,117,118]). An inflammatory local milieu within an overall lymphopenic environment has been incriminated for the inhibition of Treg function and their cell fate conversion to effector T cells (ex-Foxp3 cells), which then secrete inflammatory cytokines, increasing the risk of disease aggravation [119]. Although the dynamic regulation of Foxp3 expression is crucial in enabling the immune system to most flexibly control pathogens under various physiological conditions, a reverted, effector phenotype of ex-FoxP3 cells and the transition from a regulatory to an inflammatory program may trigger a pathogenic autoreactive T cell immunity or cytotoxic activity with serious sequelae in the context of in vivo or adoptively transferred Tregs, respectively [117]. Hence, Treg instability and plasticity are features of great importance in the pathogenesis of immunological diseases, while they represent significant barriers to the broader clinical adaptation of Tregs.

### Stabilizing Treg Phenotype to Overcome Plasticity

Since the efficacy of Treg cell therapy is closely related to their phenotypic stability in vivo, successful Treg immunotherapy requires an inflexible phenotypic profile and sustained immunosuppressive functions against a destabilizing inflammatory microenvironment. Regulatory T cells induced ex vivo (iTregs) demonstrate functional instability over nTregs as a result of the lack of iTreg-specific epigenetic changes and in particular, DNA hypomethylation at the enhancer regions of FOXP3 and other signature genes that drive phenotypic stability [120]. iTregs display only incomplete DNA demethylation despite high Foxp3 expression. Towards maintaining Treg functional benefits in vivo, fine-tuning transcriptional and epigenetic signals and pathways is a sine qua non requirement to ensure the stability of Tregs and minimize the risk of skewing into pathogenic T cells under a highly inflammatory environment.

The forced expression of Foxp3 in TCR-Tregs has been shown to counteract the consequences of endogenous FoxP3 downmodulation, thus preventing the accumulation of effector T cells in vivo, and also to convert contaminating conventional CD4 cells into Treg-like cells displaying long-term persistence [121].

The mere expression of FoxP3 is not sufficient for iTreg generation; the Treg-specific epigenome needs to be also induced in iTreg cells, in particular, Treg-specific DNA hypomethylation. The expression of Foxp3 is regulated by combinatorial epigenetic modifications and mainly relies on the methylation status of Foxp3 gene loci, and in particular, the promoter and Treg-specific demethylated region (TSDR) in the Foxp3 gene, which become demethylated in functional Tregs [120,122,123]. Indeed, the stabilization of Foxp3 expression has been successfully achieved by the epigenetic modifications of TSDR or within the promoter and enhancer regions of the Foxp3 locus, using DNA methyltransferase (DNMT) or histone deacetylase (HDACs) inhibitors or dCas9 fused to transcriptional activators (VPR), repressors (KRAB), or histone acetyltransferases (HATs, p300) [124,125,126,127,128,129]. Targeting intronic cis-regulatory elements in the Foxp3 loci (CNS1 and CNS2) by metabolic reprogramming using small molecules such as vitamin C, a cofactor for ten-eleven-translocation (TET) enzyme mediating DNA demethylation, also conferred epigenetic modulation in a TET-2-dependent manner, leading to stable Foxp3 expression and improved suppressive Treg activity [130,131,132].

Since a variety of extrinsic and intrinsic cell factors control the epigenetic, transcriptional, translational, and post-translational regulation of Foxp3 expression (reviewed in [133,134]), their direct regulation could be another key strategy to “lock in” Foxp3 expression, and subsequently Treg stability. To further add to the list of potential targets enhancing the Foxp3 stability, super-enhancers, a cluster of highly active, cell type-specific enhancers orchestrating the expression of Foxp3 and other Treg cell lineage-defining genes, as well as chromatin organizers, such as Satb1, which plays an essential role in establishing and activating Treg-specific super-enhancers have also been identified [135,136,137].

Another approach to ensure the stabilization of Foxp3 is to protect it from its negative regulators or proteins leading to polyubiquitination and subsequent proteasomal degradation. To this end, the inhibition of Foxp3 negative regulators including the Deleted in Breast Cancer 1 (DBC1) protein or Janus kinase2 (JAK2), licensed Tregs to retain high Foxp3 expression and maintain their suppressive function in experimental models of autoimmunity and GvHD, respectively [138,139]. Likewise, in proof-of-principle studies, the short hairpin RNA (shRNA)-mediated inhibition of Stub1, a ubiquitin ligase responsible for the polyubiquitination of Foxp3, or the ectopic expression of the deubiquitinase USP7, resulted in stable or increased Foxp3 expression and enhanced Treg suppressive function even within a hostile inflammatory microenvironment [140,141].

Helios, a transcription factor expressed in a large subset of Foxp3+ Tregs, mainly in tTregs, which exhibit a more stable suppressive phenotype than pTregs and iTregs by virtue of a more stabilized epigenetic signature [142], has attracted researchers’ attention as another potential target towards enhancing Treg stability. Indeed, the ectopic expression of Helios along with Foxp3 in Tregs resulted in superior suppressive function as compared to only Foxp3- and only Helios-expressing Tregs in a murine GvHD model [143].

## 4. Inhibitory Treg Signaling by the Tumor Microenvironment

In addition to tumor and stromal cells, immune cells are an essential component of the tumor microenvironment (TME) where the tumor-immune cell interplay plays a key role in tumorigenesis. In contrast to effector T cells, NK cells, DCs, and M1 macrophages, Tregs comprise tumor-promoting immune cells in which the suppressive activity is mediated by key molecules including IL-10, TGF-β, CTLA4, and IL-35. In contrast, inflammatory signals by cytokines, like TNF-a or IL-6, can decrease Treg activity probably as a homeostatic mechanism against Treg interference with immune responses to pathogens [144,145,146,147]. In addition to these signals, Treg function can be reduced via inhibitory signals directed to their TCR, which can provide negative feedback to Treg-mediated suppression. In particular, CD4+CD25+ Treg cells have a significant defect in the phosphorylation of AKT upon TCR-mediated activation, resulting in the decreased activity of downstream effectors. This defect is tightly associated with Treg suppressive function as the TCR-independent conditional activation of exogenous AKT reversed their suppressive capacities [148]. In recent years, immune checkpoint molecules including CTLA-4, PD-1, LAG-3, TIM-3, and TIGIT have been recognized as critical mediators in the biology of TME that promote cancer progression by exerting inhibitory antitumor mechanisms, whereas immune checkpoint inhibitors (ICIs) induce impressive effector T cell antitumor immune responses [149,150,151,152,153,154,155]. However, as Tregs are known to express several immune checkpoint inhibitor targets, their numbers and function may be altered by ICI immunotherapy, thus shaking the balance between effector T cell activation and suppressive effector PD-1+ Treg cell proliferation at the tumor site. Should the effect on suppressive effector Tregs be dominant, the inhibition of anticancer immunity with uncontrollable tumor growth may occur, resulting in the development of hyperprogressive disease (HPD) [156,157]. This paradoxical acceleration of the disease in a subset of patients treated with ICIs should be promptly acknowledged and urgently managed to counteract a potentially deleterious flare-up. The presence of actively proliferating PD-1+ effector Treg cells at tumor sites has been suggested as a reliable marker for HPD and their depletion in tumor tissues as a means of treating and preventing HPD in PD-1 blockade cancer immunotherapy [156]. In addition, a PD-1 and CTLA-4 combination blockade has been shown to increase effector Teff infiltration, resulting in highly advantageous Teff-to-regulatory T-cell ratios within the tumor [158].

### Functional Treg Enhancement against Inflammatory Cytokine Signaling

Expanding knowledge on the negative signal that inflammatory cytokines exert on Tregs has triggered investigation towards establishing a more robust Treg function. One such strategy is rendering Tregs resistant to factors of the inflammatory milieu driving the negative feedback of their function, such as Protein kinase C theta (PKC-θ), an inhibitor of Tregs’ suppressive function being selectively recruited to the central supramolecular activation complex (cSMAC) region of the immunological synapse (IS) between an antigen-stimulated T cell and an antigen-presenting cell. Tregs modify the type of IS that is established between the naïve T cell and the peptide-loaded dendritic cell by inhibiting the recruitment of PKC-θ to the IS [159,160]. The blockade or silencing of PKC-θ shielded Tregs from the negative effects of cytokines associated with an inflammatory milieu and ultimately enhanced their ability to prevent autoimmune colitis, as well as restored the function of defective Tregs derived from rheumatoid arthritis patients [159]. Ex vivo-generated iTregs, in which PKC-θ was neutralized by an antibody using a synthetic, cell-penetrating peptide mimic, presented enhanced immunosuppression and stability and were highly effective in preventing GvHD in a mouse model while maintaining anti-tumor surveillance [161].

An alternative to making Tregs resistant to pro-inflammatory cytokines is to neutralize cytokines in vivo. In fact, in vivo treatment with cytokine- or cytokine receptor-targeted monoclonal antibodies, such as anti-tumor necrosis factor α (anti-TNF-α) or anti-IL-6 receptor, has been shown to neutralize inflammatory cytokines while rescued Treg function in patients with rheumatoid arthritis and kidney transplant recipients, respectively [19,162,163]. It may therefore be possible to enhance Treg cell function and improve outcomes by combining adoptive Treg transfer with monoclonal Abs targeting cytokines or cytokine receptors.

In addition, Tregs could also be engineered to self-secrete neutralizing agents like monoclonal antibodies or to express receptors counteracting extrinsic, repressive signals. Using this concept, albeit in a different context, genetically engineered, anti-tumor T cell products with integrated artificial receptors engaging transforming growth factor β (TGF-β), an inhibitor of effector T and NK cells and tumor antigen-specific cellular immunity, were capable of overcoming the TGF-β-induced tumor immune evasion, being shielded from the inhibitory effects of TGF-β or functionally empowered via the conversion of TGF-β suppressive signal to an activating signal [164,165,166,167]. A similar rationale may be adapted to generate Treg products able to “tame” their corresponding inhibitory molecules even within a hostile inflammatory environment.

Besides providing the means to balance the extrinsic microenvironmental factors, genetic engineering also offers the opportunity to manipulate Tregs to secrete anti-inflammatory cytokines and thus, acquire an intrinsic advantage. The co-expression of IL-10 as additional payload in HLA-A2 CAR-Tregs further enhanced their capacity to suppress alloresponses in vitro [168], although IL-10-overexpressing FVIII-CAR Tregs unexpectedly developed a robust effector phenotype and failed to control inhibitory immune responses in a murine model of hemophilia A [103]. TGF-beta or IL-34, both suppressive Treg-specific and tolerogenic cytokines [169], could serve as additional candidates for co-expression in CAR-Tregs in order to augment their suppressive functions. Nevertheless, in engineered receptor Tregs, the tight regulation of their signal output and the determination of activation thresholds are critical to avoid unwanted toxicity.

## 5. Tissue Homeostatic Repair

In addition to being potent immune suppressors, Tregs have recently been recognized as also expressing tissue repair/regeneration signatures [170]. Unique populations of Treg cells with a broad phenotypic and functional diversity, have been discovered in a variety of non-lymphoid tissues, including the skeletal and cardiac muscle, skin, gut, lung, liver, and the CNS [171,172,173,174,175,176]. Tissue Tregs take over in the early phase of the inflammatory response, to foster the transition to a tissue milieu that favors regeneration via promoting tissue barrier repair, the proliferation and/or differentiation of non-lymphoid cell precursors, and the tissue remodeling to dampen fibrosis or astrogliosis [172,177]. These pro-regenerative effects of tissue Tregs may result from either cell–cell contacts or paracrine effects with general or tissue-specific soluble factors.

Tissue-resident Tregs, which have been aptly characterized as “regulatory chameleons” [178], share a common FOXP3+CD4+ precursor located in lymphoid organs that undergoes definitive specialization once in the target tissue, following complex transcriptional programs. A conserved transcriptional and epigenetic signature, common in mice and humans, that defines tissue-resident Tregs was identified as BATF+CCR8+ Treg cells in peripheral blood [172]. Notably, CCR8+Tregs from healthy tissues presented multiple similarities with CCR8+ Tregs isolated from tumor sites, thus strongly implicating the contribution of these cells to the human tissue repair program in both health and disease [172]. Another highly suppressive population of Treg cells, CD161+Treg cells having an all-trans retinoic acid (ATRA)-regulated gene signature, has been identified as also mediating wound healing. These CD161+Tregs were enriched in the intestinal lamina propria, particularly in Crohn’s disease, where CD161 expression on Treg cells was induced by ATRA. CD161 was co-stimulatory, and co-ligation with the TCR-induced cytokine secretion accelerated the gut epithelial barrier healing [176].

### Promoting Tissue Homeostatic Regeneration by Treg Cells

Modifications promoting homeostatic tissue repair could enhance Treg function to restore tissue damage caused by chronic inflammation, in addition to suppressing local inflammation. Examples of such modifications include engineered Tregs to overexpress Amphiregulin (AREG), a ligand for the epidermal growth receptor, and a wound-repair factor or Cellular Communication Network Factor 3 (CCN3), a growth regulatory protein, implicated in the regeneration of various tissues, including muscle [172], demyelinated neurons, and skin [179,180,181]. Engineered AREG-producing Tregs presented an enhanced ability to polarize monocytes toward an M2-like tolerogenic phenotype, which usually drives the natural wound-healing process, suggesting that engineered Tregs may further promote tissue repair [182]. The tissue-repair capacity of human AREG+ Tregs seems to operate independently from their classical suppressive function, as TCR-induced proliferation/differentiation coincided with a progressive loss of AREG [182].

## 6. Site-Specific Treg Cell Migration

The question of whether Tregs act primarily in the draining lymph node or the target tissue has drawn conflicting conclusions in different studies; however, they collectively point to the requirement of trafficking and migration to both inflamed tissues and draining lymphoid organs for effective Treg cell function in vivo [183]. Nevertheless, the trafficking properties of Treg cells proved to be highly dynamic and only the sequential migration from blood to the inflamed tissue and then to the draining lymph nodes using a panel of trafficking molecules and chemokine receptors (CCR2/CCR4/CCR5/CCR7 and P- and E-selectins) orienting their migration, ensured efficient Treg differentiation and the full execution of their immunosuppressive function [184], in an islet allograft transplantation. By entering in a coordinated fashion, both the diseased and the priming site, Tregs may limit effector T cell migration at both sites or control their priming via releasing IL-10 and TGF-b. The two sequential stages of migration seem functionally tightly linked, as the suppressive capabilities of Treg cells became limited when one migration phase was prevented [184].

### Enhancing Treg Cell Recruitment In Vivo

To be effective, adoptively transferred Tregs must home to and mediate their function at the target tissues [185]. To this end, the manipulation of Treg cell differentiation and dynamic trafficking may be therapeutically beneficial for Treg immunotherapy.

To ensure precise trafficking to specific sites/tissues in vivo, homing-receptor-tailored Tregs orchestrating the tissue-targeted migration of adoptively transferred Tregs have been developed. Tailoring thymic Tregs to express specific homing receptors for targeted migration by ex vivo expansion in Th1-polarizing conditions induced by the addition of interferon-γ and IL-12 or retinoic acid, generated epigenetically stable Tregs under prolonged exposure to inflammatory conditions, that were directed towards Th1-inflammation sites or the gut, respectively [186].

Alternatively, tissue-directed Treg recruitment was achieved by the controlled release of the C-C-Motif Chemokine 22 (CCL22) through microparticle formulations enabling the preferential recruitment of CCR4-expressing Tregs to a local site in vivo [187]. This microparticle-based system prolonged hindlimb allograft survival and promoted donor-specific tolerance [188].

## 7. Survival and Persistence

The limited to-date efficacy of adoptive immunotherapy with Tregs is, at least in part, attributed to their poor in vivo persistence. Both apoptosis and the loss of proliferation advantage could be incriminated for the observed poor persistence. Indeed, due to low Bcl-2 expression or induced oxidative stress, freshly isolated CD4+CD25+ Tregs were prone to apoptosis as compared to their CD25- counterparts or activated Tregs were driven to apoptosis upon encountering a specific antigen, respectively [189,190]. Moreover, in contrast to the well-described phenomenon of exhaustion with Tcon cells resulting from chronic stimulation and leading to poor in vivo T cell performance, Treg susceptibility to exhaustion remained an outstanding question, although the repetitive cycles of stimulation and prolonged culture required for Treg expansion were expected to affect their phenotypes, functionality, and fitness [191]. Indeed, the repetitive TCR-driven stimulation and prolonged ex vivo expansion were shown to be associated with epigenetic remodeling at loci important for Treg function and identity, including the promoter hypomethylation of genes known to downregulate T cell activation with the concomitant promoter hypermethylation of genes positively regulating TCR signaling and strong promoter hypomethylation in genes implicated in Tcon cell exhaustion [192], thus posing a risk for functional Treg exhaustion similar to what was previously reported for effector T cells [193].

To address the issue of potential Treg susceptibility to exhaustion, as it occurs with Tcons, and overcome the limitations posed by Treg capacity to normally express exhaustion-related inhibitory receptors which are often associated with enhanced Treg suppressive potential [194,195,196], Lamarche et al. used a model of tonic-signaling CAR to ask whether exhaustion has the potential to limit the in vivo efficacy of Tregs. This recent study revealed for the first time that Tregs can develop a functional deficit consistent with the concept of exhaustion, acquiring phenotypic, functional, and epigenetic changes accompanied by the complete loss of their suppressive function in vivo [197]. Therefore, Treg susceptibility to chronic stimulation-driven dysfunction must be considered and mitigated as we move forward with sophisticated adoptive Treg cellular therapies.

### Improving Treg Survival and Persistence

Given that IL-2 is indispensable, yet not Treg secretable, for Treg development in the thymus and survival in the periphery, one strategy to exploit the high sensitivity of Tregs to this cytokine and expand Treg numbers in vivo while avoiding the activation of Tcons, is by using low doses of IL-2 after adoptive Treg transfer [198,199]. The low, in contrast to high, IL-2 doses are not associated with toxicity, while they can safely expand endogenous Tregs in various disease contexts [200,201,202,203,204,205,206]. The high sensitivity of Tregs to very low IL-2 doses, based on a reduced IL-2 signaling threshold compared to effector cells, is attributed to the constitutive expression of high-affinity IL-2 receptor α chain (CD25) in Tregs, in stark contrast to intermediate affinity CD25 expressed in antigen-experienced effector cells [207]. Despite the expansion of circulating Tregs and the promising clinical results in early trials of hepatitis C virus-induced vasculitis [202], GvHD [200,208], T1D [12], SLE [209,210], and alopecia areata [211], low IL-2 monotherapy in double-blind, placebo-controlled trials was not sufficient to provide clinically relevant improvements [212,213], nor promote liver allograft tolerance [214].

Therefore, several groups leveraged the ability to selectively increase immunosuppressive Tregs via the high-affinity IL-2Rαβγc using the combination of the adoptive transfer of Tregs with IL-2 administration in vivo (Figure 2). In a nonhuman primate model, adding low-dose IL-2 to rapamycin in a setting of clinically relevant immunosuppression doubled the number of circulating Tregs and logarithmically prolonged the persistence of adoptively transferred ex vivo-expanded Tregs, which resulted in transcriptomic similarity to endogenous resting Tregs with increasing time after transfer [215]. Nevertheless, in patients with T1D or skin allografted mice, the low-dose IL-2 treatment post the adoptive transfer of polyclonal Tregs although it increased the frequency of circulating Tregs, led to only limited therapeutic benefit [41,216], probably due to the inferior in vivo performance of polyclonal Tregs over antigen-specific Tregs, as discussed earlier. In fact, when IL-2 was combined with donor-specific Tregs, but not with polyclonal Tregs, it preferentially enhanced the proliferation of the allospecific Tregs and a synergistic effect in prolonging skin allograft survival was observed [216]. Nevertheless, IL-2 receptor complexes are also expressed on immune cells other than CD4+ T cells, making them responsive to IL-2. Therefore, low IL-2 dosing may come at the expense of the activation of CD8+ and NK cells. In a clinical trial assessing the efficacy of low IL-2 to suppress allospecific immune responses and allow the complete discontinuation of maintenance immunosuppression in liver transplant recipients, rejection episodes were reported for four of five participants who initiated immunosuppression withdrawal [214]. Interestingly, exogenous IL-2, even at low doses, has been shown to induce conflicting effects on Tregs in the allo-HCT setting depending on the immune environment of the host; in a mild inflammatory state, low IL-2 regulated Treg homeostasis and suppressed GvHD, whereas in an intense inflammatory environment, the same IL-2 doses enhanced activated T cells rather than Tregs and exacerbated GvHD in a mouse model [217].

To overcome the pleiotropy of IL-2 leading to the simultaneous stimulation and suppression of immune responses as well as systemic toxicity and to specifically target transferred Tregs without activating other immune cells, Garcia’s group engineered orthogonal IL-2/IL-2 receptor (IL-2R) pairs that interact with one another, but do not interact with the natural IL-2 or IL-2R counterparts, thereby enabling the selective stimulation of target cells in vivo [218]. Following the adoptive transfer of Tregs incorporating an orthogonal IL-2R into a murine mixed hematopoietic chimerism model, orthogonal IL-2 injection selectively promoted ortho IL-2Rβ+ Treg cell proliferation without increasing other T cell subsets and facilitated donor hematopoietic cell engraftment followed by heart transplantation tolerance [219]. Likewise, in a murine major histocompatibility complex-disparate GVHD model, this approach led to enhanced GVHD survival, the in vivo selective expansion of Tregs, and importantly, the maintenance of graft-versus-tumor (GVT) responses, whereas the adoptive transfer of ortho-hIL-2Rβ+ CAR T cells into immunodeficient mice bearing CD19+Nalm6 leukemia xenografts in combination with ortho-hIL-2 administration led to 1000-fold ortho-hIL-2Rβ+CAR T expansion and rescued the antileukemic effect of an otherwise suboptimal CAR T cell dose [220,221].

Another IL-2 mutein therapeutic approach currently being evaluated in clinical trials (NCT3451422) [222], makes use of an IL-2 Fc fusion protein (Efavaleukin alfa), in which an introduced mutation decreases binding to IL-2Rβ and increases dependence to IL-2α (CD25). This preferential binding to the high-affinity IL-2R leads to enhanced cell surface retention and selective Treg signaling over recombinant IL-2. Similarly, human cytokine/antibody fusion proteins introduced into Tregs conferred protection in mouse models of colitis and checkpoint inhibitor-induced diabetes mellitus [223], while CAR-Tregs bearing membrane-associated IL-2 (mbIL-2) showed superior activity compared to control CAR-Tregs in a preclinical humanized mouse model [224]. Though promising, the efficacy of those IL-2 Treg mutants remains to be determined in clinical trials.

Other molecules known to also expand Tregs such as TNFRSF25 agonistic antibody, intravenous immunoglobulin, rapamycin, and cytokine-targeted antibodies [14,15,16,20,21,22] could be used to in vivo boost adoptively transferred Tregs in the recipient (Figure 2). Alternatively, such molecules could be administered to the donor to increase Tregs’ numbers and their potential, prior to ex vivo expansion, as it has been promisingly shown in animal models [225,226,227,228]; however, such an approach raises ethical concerns and clinical translation seems rather unrealistic.

Moreover, targeting the downstream IL-2 signaling, for instance, using signal transducer and activator of transcription 5 (STAT5)-transduced Tregs, may result in the disruption of Treg dependency on IL-2 and sustained Foxp3 expression, thus ensuring Tregs’ long-term persistence [229]. Indeed, modulating the Th2 cytokine production in vivo through STAT5 overexpression in transgenic CD4+ cells resulted in more efficient Treg expansion in vivo and reduced GvHD lethality compared to wild type Tregs in an in vivo relevant model [230]. These data implicate that the upregulation of constitutively active forms of STAT5 in Tregs by either pharmacological methods or genetic engineering could prove useful in preventing or controlling GvHD or autoimmunity.

In the context of transplantation, immunosuppression, although a sine qua non for the prevention or treatment of GvHD and graft rejection, severely compromises the endogenous or any potential adoptively transferred T cell immunity. Hence, by making Tregs resistant to specific immunosuppressive agents, they may acquire a survival advantage and remain functional even under the unfavorable conditions of intense immunosuppression (Figure 2). In the setting of adoptive T cell therapy with virus-specific T cells, our group has developed steroid-resistant, pathogen-specific T cells by the CRISPR/CAS9 genetic disruption of the glucocorticoid receptor, and other groups have also generated specific T cells with engineered resistance to various immunosuppressive agents [231] (reviewed in [232]). We foresee that this approach could be adopted for the generation of immunosuppression-resistant Treg cell products, which after adoptive transfer into transplanted patients, could remain functional and effective, thus broadening the applicability of immunotherapy with Tregs.

## 8. Treg Safety Considerations

Treg cell immunotherapy products consist of “living drugs” with potential long-term dynamics and as such, safety is of utmost importance in moving this therapy to the bedside. The non-targeted specificity of polyclonal Tregs, phenotypic instability, and potential plasticity may lead to unwanted and even deleterious effects. Genetically-modified Tregs bear additional risks associated with genetic engineering, including genotoxicity, off-tumor/on-target toxicity, and hyperactivation syndromes.

### Optimizing Treg Safety

As discussed, polyclonal, non-specific Treg cell products might entail safety risks due to their potential to lead to systemic, off-target immunosuppression and therefore, suboptimal immune responses against opportunistic infections and possibly cancer development. The use of antigen-specific Tregs, described in detail above, could decrease this risk, however, even with enriched antigen-specific Treg products, an unintentional contamination of Treg cell products with effector Tcons or a potential phenotype switching in vivo, can lead to unwanted immune responses with severe consequences. The prevention of contamination with Tcons during Treg manufacturing, ideally by magnetic-activated or fluorescence-activated cell sorting purification as a last production step, could remove the undesired Tcons, at the expense, however, of some loss of an already limited Treg population. The isolation of antigen-specific T cells expressing Treg markers, along with an in-depth assessment of their regulatory activity and ability to produce immunosuppressive factors upon antigen-specific challenge, will enable the release of safer cell products. To better control the Treg in vivo plasticity risk and further improve the safety of adoptive immunotherapy with antigen-specific Tregs, the strategies mentioned above enhancing the stability of Tregs, such as enforced Foxp3 expression [121], can be applied.

Genetically-modified Tregs having enhanced activation and expansion capabilities, may be advantageous over conventional Treg therapy against autoimmunity and allo-responses due to their specificity and potency. Notwithstanding the reported long-term safety of retro- or lentiviral-based CAR T cells in thousands of patients receiving CAR-transduced Tcons [233,234], genetic modification per se creates legitimate concerns regarding the risk of genotoxicity and insertional mutagenesis that could also apply to genetically-modified Tregs [235,236,237,238,239,240]. Inducible suicide genes or chromatin insulators have been proposed as potential means to generate safer viral vectors and consequently safer gene and cell therapy [241,242,243]. Should the employment of (viral or non-viral) delivery methods in nuclease-based genome editing with CRISPR/Cas9 or TALENs be concerned, it is vital to minimize potential undesirable, off-target mutations by performing extended in silico, in vitro, and in vivo off-target analysis which may help limit the introduction of unintended genetic changes that could affect the safety and efficacy of engineered cells and by validating precision targeting [244,245,246,247].

Although the potential of gene-modified Tcon cells to elicit off-tumor/on-target toxicity, CRS, or ICANS is well-recognized and could also apply to CAR-Treg immunotherapy, the risk of these toxicities being observed with gene-engineered CAR Tregs (should they not be contaminated with large numbers of Tcon cells or not be profoundly unstable in vivo) is, at least theoretically, substantially lower; this is due to the Treg capacities to counteract the T-effector cell function in vivo, disfavor macrophage activation, and potentially prevent the cytokine storm [248,249,250].

Towards mitigating CAR Tcon or TCR Tcon cells’ toxicity, various safety features have been incorporated in CAR-Tcons, including suicide genes (RQR8, huEGFRt, HSV-tk, iCasp9) that make the cells highly vulnerable to lysis by monoclonal antibodies or small molecules administered on demand, enabling the deletion of cells in case of severe adverse events [241,242,251].

The use of antigen-specific TCR-Tregs is associated with the risk of the potential mismatched pairing of the transgenic with the endogenous TCRs that may lead to impaired transgenic TCR expression, or in the worst-case scenario, undesirable, and even dangerous off-target effects (as reviewed in [252]). Strategies including the deletion or silencing of endogenous TCR by gene editing or RNA interference, respectively, or the cysteine modification of the transgenic TCR, making the unproductive mispairing with the endogenous TCR unlikely [253,254,255,256,257], have been proposed to prevent TCR mispairing in cancer therapy with TCR-engineered Tcons and could also be considered in TCR-Treg immunotherapy.

The potential immunogenicity of CARs induced by the presence of non-human sequences (scFvs) in the CAR construct, other components of the CAR-T, or the presence of residual viral or other non-human origin proteins during CAR T cell manufacturing represents another safety challenge to consider regarding CAR-Tregs [258]. Humanized alloantigen-specific CAR (A2-CAR)-Tregs to annihilate the risk of undesirable immune responses have been developed and shown to be effective in suppressing HLA-A2+ cell-mediated xenogeneic GvHD and diminish the rejection of human HLA-A2+ skin allografts [259].

## 9. Conclusions

The feasibility and safety of adoptive immunotherapy with Tregs have been demonstrated in pivotal clinical studies that have suggested Treg cell therapy as a promising therapeutic option for patients suffering from autoimmune diseases or the immunological complications of hematopoietic cell or solid organ transplantation (Table 2). Independent of their suppressive activity, a new role for Tregs has been recently recognized with regard to promoting tissue repair and wound healing, thus opening the potential for also treating non-autoimmune disorders with Tregs.

Nevertheless, the efficacy of Treg adoptive immunotherapy to reshape the immune balance toward a specific and long-lasting tolerance still faces a plethora of challenges and the establishment of immune tolerance remains rather elusive. In the recent two decades, the dramatic progress in our understanding of basic Treg cell biology and its association with the development of autoimmunity or allo-responses in transplantation, along with the advent of new genetic engineering tools, has led to the concept and development of “designer” Tregs towards enhancing the potency, long-lasting effect, and safety of this tolerogenic therapy [260]. Unequivocally, genetically-modified Tregs have attracted increasing scientific, as well as commercial attention. An important step further, towards the broader applicability of adoptive Treg cell therapy especially in the autoimmunity context, where autologous and possibly suboptimally performing ex vivo and in vivo cells are being used, will be the development of Treg biobanks with universal, off-the-shelf products (Figure 3). Such universal Tregs will enable the use of a single batch of a Treg cell product to treat multiple patients while minimizing the manufacturing time and cost of those living drugs. In this case, it would be ideal, in order to escape immune recognition by the host, to use either HLA-deficient, gene-edited Tregs or Tregs having the non-classical HLA-E or HLA-G either ectopically expressed or epigenetically reprogrammed, to also bypass the NK cell-mediating killing [56,261].

The numerous gene engineering tools and the smart strategies for redirecting cell specificity that are currently available or under development, along with new semi- or fully automated systems for manufacturing have generated an exciting, yet challenging, new era in Treg adoptive immunotherapy [262] towards finding the proper balance between immune tolerance and immune surveillance, thus bringing us closer to definitive treatments for difficult-to-cure diseases.

## Figures and Tables

**Figure 1 cancers-15-05877-f001:**
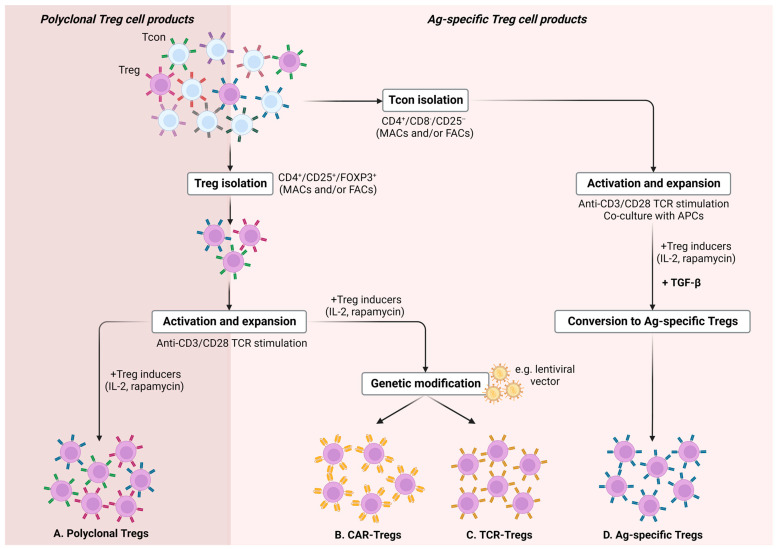
Strategies to ex vivo generate Treg cells for adoptive immunotherapy. (**A**) A polyclonal Treg cell population can be generated after isolation of Tregs, CD3/CD28 activation, and ex vivo expansion in the presence of high-dose IL-2 or other Τreg inducers. (B,D) Antigen-specific Treg cell products can be produced either by genetically modifying isolated polyclonal Treg cells to express a chimeric antigen receptor (CAR-Tregs) (**B**) or an artificial T cell receptor (TCR-Tregs) targeting a disease-relevant antigen of interest (**C**) or by converting antigen-specific Tcons to antigen-specific Tregs, e.g., via culture in the presence of TGF-ß (**D**). Tcon: T conventional cells, Treg: T regulatory cells, Ag: antigen, CAR: chimeric antigen receptor, TCR: T cell receptor, APCs: antigen-presenting cells, IL-2: interleukin-2, MACs: Magnetic Cell Separation, FACs: fluorescence-activated cell sorting, TGF-ß: transforming growth factor beta. Created with BioRender.com (accessed on 7 December 2023).

**Figure 2 cancers-15-05877-f002:**
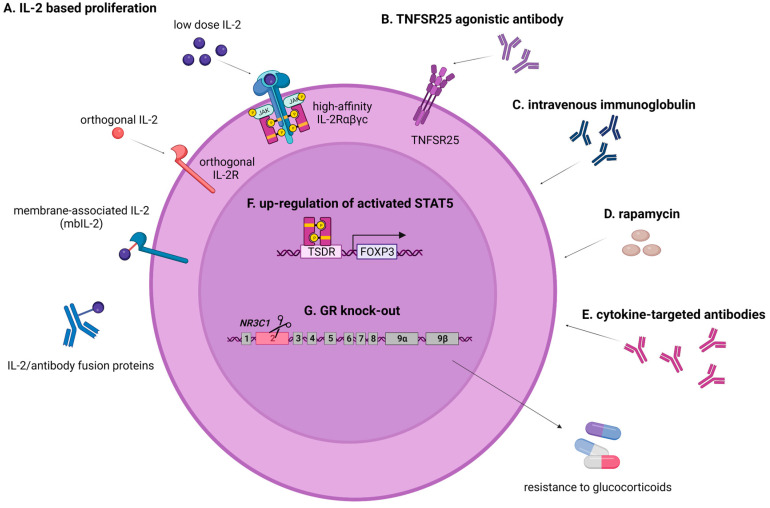
Strategies to enhance the survival and persistence of adoptively transferred Tregs. The figure depicts different approaches to enhance the in vivo survival and persistence of Tregs after adoptive Treg transfer; these include administration of low-dose IL-2 or its mutants for selective in vivo stimulation of Tregs and not other immune cells (**A**), administration of other molecules that can also in vivo boost adoptively transferred Tregs such as TNFSR25 agonistic antibody (**B**), immunoglobulin (**C**), rapamycin (**D**), and cytokine-targeted antibodies (**E**), upregulation of STAT5 for sustained, IL-2-independent Foxp3 expression, and (**F**) knock-out of the glucocorticoid receptor (GR) to render Tregs resistant to glucocorticoids (**G**). Created with BioRender.com (accessed on 7 December 2023).

**Figure 3 cancers-15-05877-f003:**
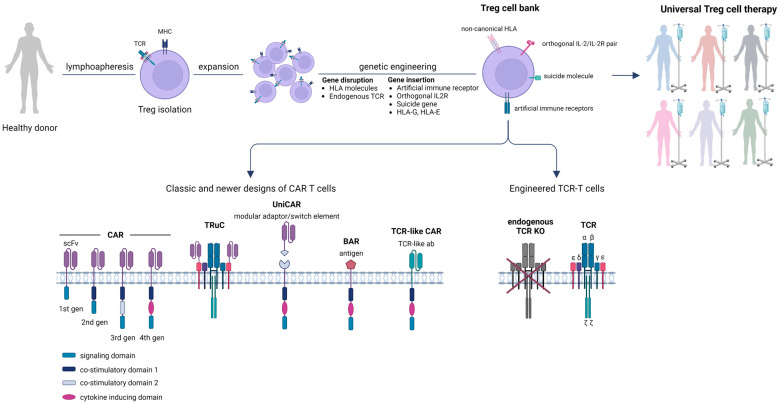
Universal or “one-size-fits-all” Treg cell therapeutics. Universal Treg cell therapy will be based on the development of biobanks of healthy donor-derived, off-the-shelf Treg cell products generated by multiplex genetic engineering to overcome the HLA barriers by disrupting the HLA molecules or knocking in non-classical HLAs (HLA-E/-G) and to express an artificial disease-relevant immune receptor—instead of their native TCR—for antigen specificity (classic and newer designer CARs or transgenic TCRs), an orthogonal IL2/IL-2R pair for enhanced in vivo persistence and/or a suicide gene as a safety switch. Treg: T regulatory cells, TCR: T cell receptor, MHC: major histocompatibility complex, HLA: human leukocyte antigen, IL-2: interleukin-2, IL-2R: interleukin-2 receptor, HLA-G: human leukocyte antigen G, HLA-E: human leukocyte antigen E, CAR: chimeric antigen receptor, scFv: single-chain variable fragment, gen: generation, TRuC: TCR fusion construct, UniCAR: universal CAR-T cells, BAR: chimeric B-cell antibody receptor, ab: antibody, KO: knockout. Created with BioRender.com (accessed on 7 December 2023).

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
