# Peer review of "Promises and Pitfalls of Next-Generation Treg Adoptive Immunotherapy"

_cancers, 2023, doi:10.3390/cancers15245877_

Round 1

Reviewer 1 Report

Comments and Suggestions for Authors

Panayiota Christofi and co-authors present a high quality and well-written review manuscript focused on promises and pitfalls towards next generation Treg adoptive immunotherapy.

Authors present a research that discusses the current achievements and existing challenges towards clinically translating Tregs into a living drug therapy for a variety of inflammatory conditions.

Authors suggest that the feasibility and safety of adoptive immunotherapy with Tregs has been demonstrated in pivotal clinical studies that suggested Treg cell therapy as a promising therapeutic option for patients suffering from autoimmune diseases or the immunological complications of hematopoietic cell- or solid organ-transplantation. Independent of their suppressive activity, a new role for Tregs has been recently recognized with regard to pro moting tissue repair and wound healing, thus opening the potential for also treating non- autoimmune disorders with Tregs.

Authors cover such aspects as:

- Specificity of tolerance

- Treg functional stability versus plasticity

- Inhibitory Treg signaling by the tumor microenvironment

- Tissue homeostatic repair

- Site-specific Treg cell migration

- Survival and persistence

- Treg safety considerations

Finally, authors conclude that the efficacy of Treg adoptive immunotherapy to reshape the immune balance toward a specific and long-lasting tolerance still faces a plethora of challenges and the establishment of immune tolerance remains rather elusive. Authors also suggest that the numerous tools and the smart strategies that are currently available or under development, have generated an exciting, yet challenging, new era in Treg adoptive immunotherapy, towards finding the proper balance between immune tolerance and immune surveillance, thus bringing us closer to definitive treatments for difficult to cure diseases.

Overall, the manuscript is highly valuable for the scientific community and should be accepted for publication.

======================

Other comments to authors:

1) Please check for typos throughout the manuscript.

2) Please improve figures/tables where appropriate.

3) With regards to manufacturing of CAR-T and T cell products – authors are kindly encouraged to cite the following article that describes the recent advances in the development of bioreactors for manufacturing of adoptive cell immunotherapies. DOI: 10.3390/bioengineering9120808

Author Response

Thank you for your constructive and insightful feedback on our manuscript. Typos have been checked throughout the manuscript, Figure 1 has been refined, and two new tables have been added to the revised manuscript—one outlining the phenotypic and functional characteristics of different Treg subsets (Table 1) and another summarizing the status and main points of clinical trials featuring Treg adoptive immunotherapy (Table 2). Finally, the recommended reference from the reviewer has also been included in the revised manuscript, line 766 which now reads as: “The numerous gene engineering tools and the smart strategies for redirecting cell specificity that are currently available or under development, along with new semi- or fully-automated systems for manufacturing have generated an exciting, yet challenging, new era in Treg adoptive immunotherapy….”

Reviewer 2 Report

Comments and Suggestions for Authors

The aim of the article by Christofi et al. is to critically review the advantages and disadvantages of Treg cell-based therapy.

In the introduction, the origin, classification, and characteristics of Tregs are presented in detail and in an understandable way. They also provide a concise summary of the trends in Treg-based cell therapies.
The theoretical basis and limitations of polyclonal Treg therapy are also summarized and used to illustrate the features, advantages, and limitations of antigen-specific Treg therapy. The possibilities of ex vivo Treg-based adoptive immunotherapy, CAR Treg, TCR Treg, and Ag-specific Treg training methods are presented in detail and in a clear and understandable manner. 
Since Treg instability and plasticity are important features in the development of immunological diseases and represent a major obstacle to the wider clinical adaptation of Tregs, aspects of Treg phenotype stabilization are also detailed. They also highlight why this is important in relation to the maintenance of plasticity. They also carefully consider the crucial fact that Tregs express a variety of immune checkpoint inhibitor targets; ICI therapy may change their number and function, impairing Treg function within the TME and speeding up tumor progression. For this reason, they present the principles and methodology of functional Treg amplification against inflammatory signals. A separate chapter is also devoted to a critical summary of the role of Tregs in tissue repair and its therapeutic potential. The therapeutic advantages of recruiting Tregs and manipulating Treg differentiation and dynamic movement for this purpose are also detailed. Aspects of Treg's survival are then presented in great detail and critically.
Finally, the safety aspects of Treg-based cell therapies are evaluated in a very didactic manner.

This review article is valuable, well thought out, and well structured. The use of English is correct, only minor language polishing is required. The illustrations are illustrative and help to understand what is written. The literature used is up-to-date and adequate.
Not only is the topic summarized, but also forward-looking research and therapeutic options are formulated.

Author Response

We sincerely appreciate your thorough and positive evaluation of our article. We are grateful for your detailed feedback, and we are pleased to know that the content and presentation of the manuscript have been well-received.

Reviewer 3 Report

Comments and Suggestions for Authors

The authors provide a very extensive summary of the relatively new topic of Tregs in immunotherapy. Overall, very informative and though discussion on the subject. Some of the comments:

TCON acronym is not initially defined. Figure 1 can be more explanatory. How does CD28 ets play a role in the specific tregs such as antigen specific and tcr specific ones.

The authors can provide a table for various tregs and their surface markers for identification. It would be nice to have the same for clinical trials or any publications that have shown promising mouse model data.

The paragraph line 74-77 seems out of place. Could the authors clarify more on this?

The references should be at the end of the sentences rather than middle. Could the authors provide any specific references for universal tregs. Have any studies been conducted even in model systems?

Author Response

Thank you for your constructive and insightful feedback on our manuscript. Tcon acronym is defined in line 191 in the revised manuscript. Following reviewer suggestions, Figure 1 has been refined, and two new tables have been added to the revised manuscript—one outlining the phenotypic and functional characteristics of different Treg subsets (Table 1) and another summarizing the status and main points of clinical trials featuring Treg adoptive immunotherapy (Table 2). In addition, the paragraph from lines 74 to 77 has been revised in the manuscript which now reads as: Tregs reshape immune responses with precision, executing their regulatory function in a sophisticated and tailored manner as opposed to the conventional, general immunosuppressive approaches. This precise immune regulation, particularly in contexts like autoimmunity and transplantation is of highest importance”. Furthermore, in the revised manuscript, references are appropriately positioned at the end of sentences. Finally, to our knowledge, there are only two preclinical studies and no clinical references available for UniCAR Tregs, a point that is now clarified in the revised manuscript, lines 286-289.